# Efficient Underground Target Detection of Urban Roads in Ground-Penetrating Radar Images Based on Neural Networks

**Wei Xue** [1,2,3], **Kehui Chen** [1,2,3], **Ting Li** [1,2,3], **Li Liu** [1,2,3] **and Jian Zhang** [4,*]

1   School of Automation, China University of Geosciences, Wuhan 430074, China
2   Hubei Key Laboratory of Advanced Control and Intelligent Automation for Complex Systems, Wuhan 430074, China
3   Engineering Research Center of Intelligent Technology for Geo-Exploration, Ministry of Education, Wuhan 430074, China
4   School of Electronic Information, Wuhan University, Wuhan 430072, China
*   Correspondence: zhangjian@whu.edu.cn

**Abstract:** Ground-penetrating radar (GPR) is an important nondestructive testing (NDT) tool for the underground exploration of urban roads. However, due to the large amount of GPR data, traditional manual interpretation is time-consuming and laborious. To address this problem, an efficient underground target detection method for urban roads based on neural networks is proposed in this paper. First, robust principal component analysis (RPCA) is used to suppress the clutter in the B-scan image. Then, three time-domain statistics of each A-scan signal are calculated as its features, and one backpropagation (BP) neural network is adopted to recognize A-scan signals to obtain the horizontal regions of targets. Next, the fusion and deletion (FAD) algorithm is used to further optimize the horizontal regions of targets. Finally, three time-domain statistics of each segmented A-scan signal in the horizontal regions of targets are extracted as the features, and another BP neural network is employed to recognize the segmented A-scan signals to obtain the vertical regions of targets. The proposed method is verified with both simulation and real GPR data. The experimental results show that the proposed method can effectively locate the horizontal ranges and vertical depths of underground targets for urban roads and has higher recognition accuracy and less processing time than the traditional segmentation recognition methods.

**Keywords:** ground-penetrating radar; underground target detection; urban road; neural network; robust principal component analysis; fusion and deletion algorithm

## 1. Introduction

Underground target detection plays an important role in urban road exploration and can be used to conduct road maintenance and management. Underground targets of urban roads mainly include voids, pipes, and cables. Due to its advantages of having a nondestructive nature, high scanning efficiency, and penetration, ground-penetrating radar (GPR) has been widely used in urban road exploration [1–5]. Because of the complex structures of underground targets, the interpretation of GPR data still mainly relies on skilled operators. In general, the time needed for analyzing GPR data is considerably longer than the time taken for data acquisition. When long-distance urban roads need to be surveyed, the amount of GPR data is very large and manual handling is apparently powerless. Therefore, it is necessary to develop automatic target detection methods, which can not only increase the interpretation efficiency, but also avoid the influence of subjective factors.

At present, automatic target detection methods in GPR data can be divided into two categories: machine learning methods and deep learning methods [6,7]. Automatic detection methods based on machine learning generally include three stages: preprocessing, feature extraction, and signal classification [8]. Preprocessing mainly performs clutter and

noise suppression. Feature extraction reduces the preprocessed data to form a set of measures that represent the data. Signal classification uses classifiers to recognize the object according to its features. Al-Nuaimy et al. [9] proposed a method based on a neural network to recognize buried objects in GPR B-scan images. The method employs ensemble mean subtraction to remove the background clutter and uses the Welch power spectra of A-scan signal segments as the features. However, the ensemble mean subtraction fails to remove the non-horizontal clutter, which causes some incorrect identification for sections of the targets. Wu et al. [10] presented a method based on a support vector machine (SVM) to detect holes under a railway with GPR signals. The method uses dyadic wavelet transform to extract the energy features of A-scan signals and reaches a high recognition rate, but it only recognizes the whole A-scan signal and cannot provide the depth information of the holes. Frigui et al. [11] proposed an algorithm for landmine detection based on K-nearest neighbor (KNN) using the GPR data. The method uses edge histogram descriptors (EHD) for feature extraction and obtains better performance than the hidden Markov model (HMM) because EHD applies fuzzy techniques to distinguish true detection from false alarms. Torrione et al. [12] developed a method based on the histogram of oriented gradients (HOG) features and the random forest method to detect landmines in the GPR data. The results indicate that HOG features provide better target classification performance than EHD and HMM. Xie et al. [13] used SVM to recognize RC structure voids in GPR images. The predictive deconvolution process is used to suppress clutter and three time-domain statistical features are extracted for each segmented A-scan signal. The method achieves high accuracy in depth and lateral range locations, but the accuracy is strongly affected by the noise level. Núez-Nieto et al. [14] presented a method based on neural networks to detect the landmines and unexploded ordnance (UXO) in GPR images and demonstrated that the neural network was superior to logistic regression. However, the method adopts A-scan signals within a window as the features. It also only recognizes the lateral ranges of targets and lacks the depth location of targets. Harkat et al. [15] presented a binary radial basis function (RBF) neural network based on the multiobjective genetic algorithm (MOGA) to detect targets in GPR images. The method uses high-order statistical cumulants of a segmented image with 41*41 pixels as the features and obtains better classification performance than convolutional neural network (CNN) and SVM. However, the MOGA framework is time-consuming due to the large size of the features. Recently, image target detection methods based on depth learning have gradually become more popular in the field of image recognition [16–18]. Deep learning methods learn the feature representations directly from the original image instead of traditional manual feature extraction in machine learning methods, which have high robustness in the detection task. As the most popular deep learning network, CNN uses multiple convolutional layers and pooling layers to extract features and uses the fully connected layer and the softmax layer to perform the classification. CNN has been proposed for the interpretation of GPR images such as concealed crack detection in asphalt pavement [19], rebar detection and localization in concrete [20,21], material type and shape classification and soil type classification [22], and internal defect detection in roads [23].

Though deep learning methods have gained increasing attention in GPR target detection, they still have two main limitations. One limitation is that they need a large amount of labeled data to train the deep learning models. Another limitation is that deep learning models have complex network structures, and the training and testing of deep learning models require high-performance graphical processing units (GPUs). Therefore, this research still focuses on machine learning methods. To obtain the locations of underground targets for urban roads, traditional machine learning methods need to recognize each segmented A-scan signal in the B-scan image. However, it is not necessary to identify all segmented signals for classifiers due to the sparse distribution of underground targets in B-scan images. Therefore, reducing the number of segmented signals to be recognized is a potential approach to improve detection efficiency.

According to the above analysis, an efficient method based on neural networks is proposed for underground target detection of urban roads in GPR images. The proposed

method first uses robust principal component analysis (RPCA) to eliminate the clutter in the B-scan image and then divides the target detection into whole recognition and segmentation recognition of A-scan signals in the B-scan image. The whole recognition of A-scan signals is used to obtain the horizontal regions of targets, while the segmentation recognition of A-scan signals is used to obtain the vertical regions of targets, which can decrease the recognition for segmented A-scan signals not from targets. In addition, the fusion and deletion (FAD) algorithm is used to optimize the horizontal regions of targets. The experimental results with simulation and real GPR data demonstrate the effectiveness of the proposed method in underground target detection of urban roads.

The remainder of this paper is organized as follows. Section 2 describes the proposed method in detail. Section 3 presents the experimental results and discussion. Section 4 lists the conclusion.

## 2. Theory and Method

### 2.1. Detection Model of GPR

GPR data may be composed in three different forms: A-scans, B-scans, and C-scans. The A-scan signal is presented in the form of a time-series signal, the B-scan image is constructed by stacking multiple A-scan signals, and the 3D C-scan data cube is formed by stacking multiple B-scan images. This paper mainly researches target detection in B-scan images.

A simple detection model of a buried target for GPR [24] is shown in Figure 1. When GPR antennas scan the target along the horizontal axis, the antenna position $x_n$ and the time delay $t_n$ corresponding to the target approximately satisfy the hyperbolic equation:

$$\frac{t_n^2}{t_0^2} - \frac{4(x_n - x_0)^2}{(vt_0)^2} = 1 \tag{1}$$

where $x_0$ is the horizontal position of the target, $t_0$ is the time delay at the position $x_0$, and $v$ is the wave velocity in the underground medium. Therefore, the target detection in the B-scan image mainly refers to the recognition of hyperbolic patterns.

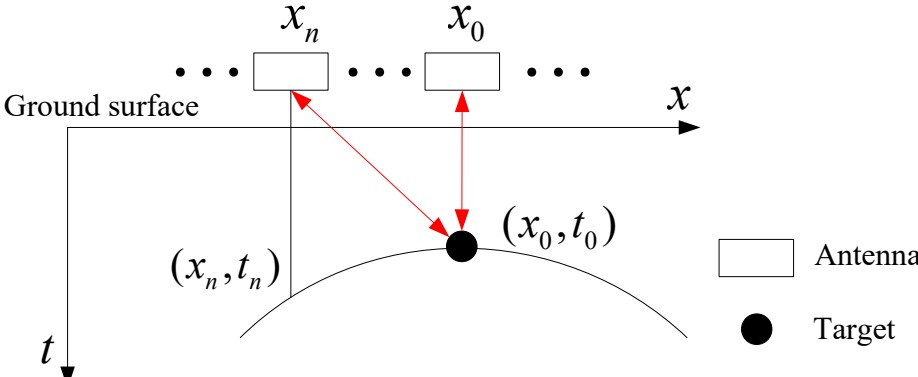

**Figure 1.** Detection model of a buried target for GPR.

### 2.2. Detection Method

A flowchart of the proposed target detection method for GPR images based on neural networks is shown in Figure 2. First, the preprocessing stage employs RPCA to suppress clutter in the original B-scan image. Second, three time-domain statistics of each A-scan signal in the image are selected to form the feature vector, and one back-propagation (BP) neural network is used to recognize the horizontal regions of targets. Third, the recognized horizontal regions are optimized by the FAD algorithm. Finally, the same three statistics of each segmented A-scan signal in horizontal regions are used to form the feature vector, and another BP neural network is used to recognize the vertical regions of targets, and then

the final target regions can be obtained from the horizontal and vertical regions. The main stages of the proposed method are described in detail in the following sections.

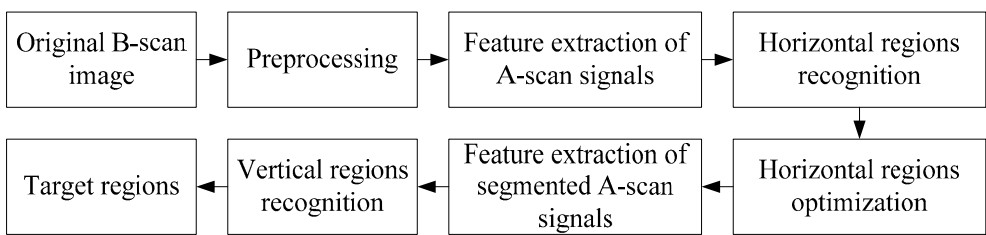

**Figure 2.** Flowchart of the proposed method.

### 2.2.1. Preprocessing

The original GPR B-scan image contains background clutter, target response, and noise. The background clutter can be caused by direct waves from the ground, reflections from the underground layer and non-targets, and multiple reflections from the target, which has a great impact on the target detection in the B-scan image. Therefore, the preprocessing stage mainly involves suppression of background clutter.

The two-dimensional GPR B-scan image can be denoted by $X = [x_1, x_2, \cdots, x_N] \in R^{M \times N}$, where M is the number of sampling points in each trace (A-scan) and N is the number of traces. The ith A-scan signal $x_i \in R^{M \times 1} (i = 1, 2, \cdots, N)$ is composed of the background clutter $a_i$, the target response $s_i$, and noise $e_i$. Therefore, the B-scan image can be expressed as [25]

$$X = A + S + E \tag{2}$$

where $A = [a_1, a_2, \cdots, a_N]$ is the low-rank clutter matrix, $S = [s_1, s_2, \cdots, s_N]$ is the sparse target response matrix, and $E = [e_1, e_2, \cdots, e_N]$ is the full-rank noise matrix. The low-rank and sparse property of the B-scan image can be utilized to discriminate the clutter A and the target response S by RPCA decomposition.

RPCA is proposed to overcome the drawback of classical principal component analysis (PCA), which is that it is sensitive to outliers. RPCA aims to find a low-rank structure in high dimensional data by solving a convex optimization problem [26]:

$$\min_{A,S} \|A\|_* + \lambda \|S\|_1 \ \ s.t. \ X = A + S \tag{3}$$

where $\| \bullet \|_*$ is the nuclear norm of the matrix argument (the sum of its singular values), $\| \bullet \|_1$ is the $l_1$-norm of the matrix (the sum of the absolute values of matrix entries), $\lambda$ is a positive regularization parameter, and s.t. is the abbreviation of "subject to".

Instead of directly solving the optimization problem in (3), an augmented Lagrangian function is constructed as follows:

$$L(A, S, Y, \alpha) = \|A\|_* + \lambda \|S\|_1 + \langle Y, X - A - S \rangle + \frac{\alpha}{2} \|X - A - S\|_F^2 \tag{4}$$

where Y is the Lagrange multiplier matrix, $\langle \bullet \rangle$ is the inner product, and $\alpha > 0$ is the penalty factor. The optimization problem can be solved by minimizing the function $L(A, S, Y, \alpha)$ with the augmented Lagrange multipliers (ALM) algorithm.

After RPCA decomposition, the sparse matrix S is used as the target image, and the low-rank matrix A is removed as the clutter.

### 2.2.2. Feature Extraction of A-Scan Signals

Feature extraction is a crucial aspect of target recognition. Here, the features extracted from the A-scan signal are used to identify the horizontal regions of targets. Features based on the A-scan signal include frequency-domain spectral features, wavelet-domain features, and time-domain features. Compared with the other types of features, time-domain feature extraction from the A-scan signal requires less computation. Time-domain features can be

extracted from the original signal, as well as from various transformations of the original signal, such as absolute value, envelope of Hilbert transform, and the short-term average to long-term average (STA/LTA) ratio [27]. The three transformations of the original signal can strengthen the description for the variation in signal energy but weaken the description for the variation in signal oscillation. For GPR, target reflections and non-target reflections can be better distinguished from the variation in signal oscillation. Therefore, considering the feature representation capability of signal oscillation and the number of features, three time domain statistics of the original A-scan signal are selected as the features, which are expressed as follows:

(1)     Mean absolute deviation

$$MAD_i = \frac{1}{M}\sum_{m=1}^{M}|x_i(m) - \overline{x}_i| \tag{5}$$

(2)     Standard deviation

$$STD_i = \sqrt{\frac{1}{M}\sum_{m=1}^{M}(x_i(m) - \overline{x}_i)^2} \tag{6}$$

(3)     Fourth root of the fourth moment

$$FRFM_i = \sqrt[4]{E(x_i - \overline{x}_i)^4} \tag{7}$$

where $x_i(m)$ is the mth element in the ith A-scan signal $x_i$, $\overline{x}_i$ is the mean value of $x_i$, and $E(\bullet)$ is the expected value function.

### 2.2.3. Target Horizontal Region Recognition

Target horizontal region recognition aims to obtain lateral ranges of targets by classifying A-scan signals. Here, the neural network is chosen as the classifier because it has high robustness to noise and is available for recognizing target reflections in A-scan signals [28].

The neural network is an information processing technology similar to the human nervous system, which can solve complex problems through nonlinear mapping [29]. The neural network consists of many basic computing neurons that are mesh-connected to each other for learning. A training algorithm is used to update the weights and biases until the actual output of the multi-layer perceptron (MLP) overlaps the desired output.

The neural network used here is a standard three-layer feedforward network trained with the backpropagation (BP) algorithm. The architecture of the BP neutral network is shown in Figure 3. The input layer consists of three neurons (one neuron per feature), and the output layer contains two neurons (target reflection or non-target reflection). The number of neurons in the hidden layer is basically of the same order of magnitude as the mean term of the proportion between the number of neurons in the input and output layers [30]. By means of the least-squares method, the number of neurons in the hidden layer is determined approximately by the empirical formula:

$$l < \sqrt{(p+q)} + a \tag{8}$$

where $p$ is the number of neurons in the input layer, $q$ is the number of neurons in the output layer, and $a$ is constant with $0 < a < 10$. Here, the number of neurons in the hidden layer is set to 10 according to several simulations, which can provide high accuracy and low computational complexity.

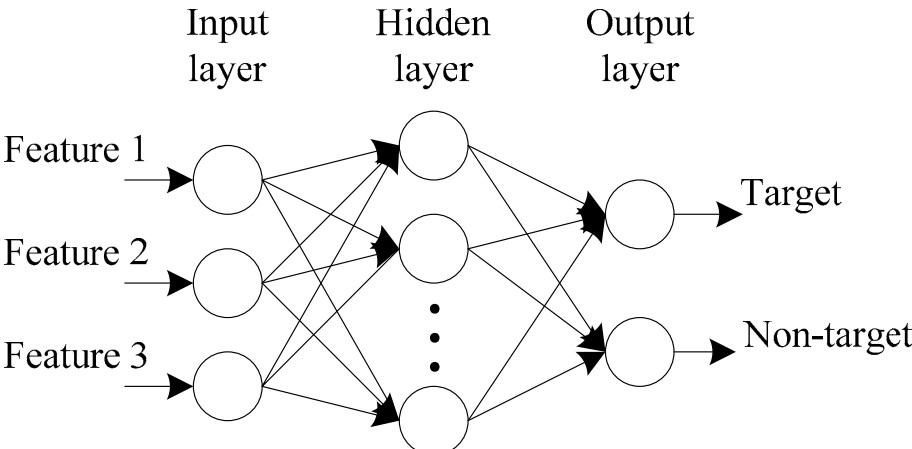

**Figure 3.** BP neural network architecture.

In the BP neural network, the output of the *j*th neuron in the *k*th layer is given by

$$y_{k,\,j} = f\left(\sum_{i=1}^{L_{k-1}} w_{ij} y_{k-1,\,i} + b_{k,\,j}\right) \tag{9}$$

where $j = 1, 2, \cdots, L_k$ and $L_k$ is the number of neurons in the *k*th layer, $f(\bullet)$ is the activation function, $w_{ij}$ is the weight, and $b_{k,\,j}$ is the bias.

The activation functions mainly include the logsig function, tansig function, and purelin function. The first two are nonlinear functions, and the latter is a linear function. Here, the hidden layer adopts the logsig function and the output layer adopts the purelin function.

In the BP algorithm, the gradient of the error function with respect to each weight is computed, and the weights are adjusted along the downhill direction of the gradient in order to reduce the error. Generally, such a learning scheme is slow, so a momentum term is introduced to increase the convergence rate [31]. The weight adjustment with a momentum term can be expressed as follows:

$$\Delta w_{ij}(n) = -\eta \frac{\partial E(w)}{\partial w_{ij}(n)} + \alpha \Delta w_{ij}(n-1) \tag{10}$$

where *n* is the index of iterations, $\eta$ is the learning rate, $\frac{\partial E(w)}{\partial w_{ij}(n)}$ is the gradient of the error function with respect to the weight, and $\alpha$ is the momentum factor.

Based on the BP neural network designed above, the steps of target horizontal regions recognition are as follows:

1. Training set construction. Km1 A-scan signals with target reflections and kn1 A-scan signals without target reflections are selected from the B-scan images for training, and three features of each selected A-scan signal are extracted. Then, the features are normalized to construct the training set TR1, including km1 positive samples and kn1 negative samples. The output of the positive sample is set to [1 0], and the output of the negative sample is set to [0 1].

2. Network training. The training set TR1 is used to train the designed BP neural network, and the network model NET1 is obtained.

3. Horizontal region recognition. The three features of all A-scan signals in the test B-scan image are extracted and normalized to construct the test set TE1. Then, the model NET1 is used to classify the samples in TE1. Assuming that *K1* samples are identified as positive samples, the corresponding A-scan signals can be written as $x_i$ $(i = i_1, i_2, \cdots, i_{K1})$. Then, the target horizontal regions can be denoted as $H1 = \{i_k \Delta d,\ 1 \leq k \leq K1\}$, where $\Delta d$ is the trace interval.

### 2.2.4. Optimization of Target Horizontal Regions

Due to the influence of residual clutter and target reflection amplitude fluctuation, false detection and missing detection are inevitable in target horizontal region recognition. In general, the projection of the target reflection hyperbola in the horizontal direction should be continuous and have a certain width. Based on the characteristics of target reflections, a fusion and deletion (FAD) algorithm is proposed to optimize the target horizontal regions.

The illustration of target horizontal region recognition is shown in Figure 4. The recognized target A-scan signals are marked in black, and the recognized non-target A-scan signals are marked in white. For the target horizontal regions $H1 = \{i_k \Delta d, 1 \leq k \leq K1\}$, the steps of the FAD algorithm are described as follows:

1. Fusion processing. Fusion processing refers to further judgment for non-target A-scan signals between two adjacent target regions, which aims to reduce the false negative rate (missing detection). The judgment can be expressed as

$$
\begin{cases}
x_i \text{ is the target signal,} & 1 < i_{k+1} - i_k \leq dth/\Delta d \\
x_i \text{ is the non} - target\ signal, & else
\end{cases}
\tag{11}
$$

where $x_i\ (i_k+1 \leq i \leq i_{k+1}-1)$ is the non-target A-scan signal between two discontinuous target A-scan signals $x_{i_k}$ and $x_{i_{k+1}}$ in $H1$ and *dth* is the horizontal interval threshold. Through the processing, the two adjacent target regions with intervals less than *dth* will be fused together. Then, the horizontal region after fusion processing can be described as $H2 = \{i_k \Delta d, 1 \leq k \leq K2\}$, where $K2 \geq K1$.

2. Deletion processing. Deletion processing further judges the target horizontal regions after fusion processing, which aims to decrease the false positive rate (false detection). The judgment can be represented as

$$
\begin{cases}
x_i \text{ is the non} - target\ signal, & i_k - i_1 = k - 1 \leq wth/\Delta d \\
x_i \text{ is the target signal,} & else
\end{cases}
\tag{12}
$$

where $x_i\ (i_1 \leq i \leq i_k)$ is the continuous target A-scan signal in one target region in $H2$ and *wth* is the horizontal width threshold. Through the processing, the target region with width less than *wth* in $H2$ will be deleted. If $i_k = i_1$, $x_{i_k}$ is an isolated target signal, and the location point $i_k$ will also be deleted. Then, the final optimized horizontal regions after fusion and deletion processing can be denoted as $H3 = \{i_k \Delta d, 1 \leq k \leq K3\}$, where $K3 \leq K2$.

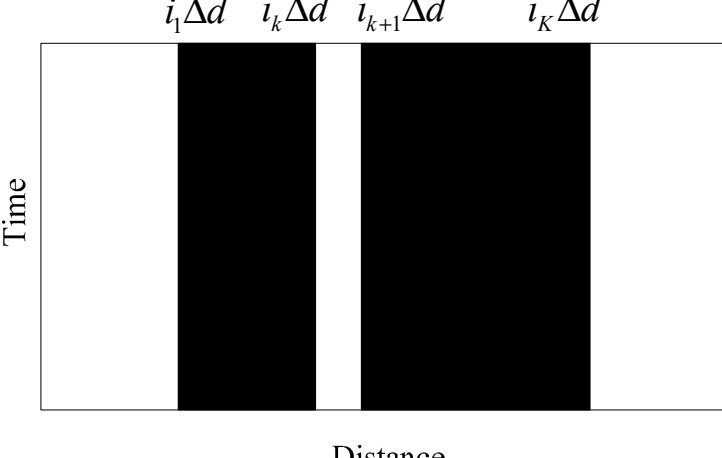

**Figure 4.** Illustration of target horizontal region recognition.

### 2.2.5. Feature Extraction of Segmented A-Scan Signals

After the target horizontal regions are obtained from the recognition of A-scan signals, the vertical regions of targets need to be further determined from the recognition of segmented A-scan signals in the horizontal regions *H*3. The segmentation of one A-scan signal is shown in Figure 5. Assuming that the segment length is *ml* and the time-sampling interval is $\Delta t$, the corresponding number of segments in one A-scan signal can be written as:

$$KL = floor\left(\frac{M}{ml}\right) \tag{13}$$

where $floor(\bullet)$ is the round down function. The basic selection principle of segment length *ml* is that the length should be as short as possible and the segment can contain enough information about the target reflection. Generally, the range of *ml* is $\left[\frac{0.5}{f_c \Delta t}, \frac{1.5}{f_c \Delta t}\right]$, where $f_c$ is the antenna central frequency.

**Figure 5.** Segmentation of one A-scan signal.

Then, the segmented A-scan signal can be expressed as

$$xs_{i,r}(j) = x_i((r-1) \bullet ml + j) \tag{14}$$

where $i = i_1, i_2, \cdots, i_{K3}$, $r = 1, 2, \cdots, KL$, and $j = 1, 2, \cdots, ml$.

Similar to feature extraction of the A-scan signal, the same three time domain statistics of the segmented A-scan signal are chosen as the features, which are expressed as:

(1)  Mean absolute deviation

$$MAD_{i,r} = \frac{1}{ml} \sum_{j=1}^{ml} |xs_{i,r}(j) - x\bar{s}_{i,r}| \tag{15}$$

(2)  Standard deviation

$$STD_{i,r} = \sqrt{\frac{1}{ml} \sum_{j=1}^{ml} (xs_{i,r}(j) - x\bar{s}_{i,r})^2} \tag{16}$$

(3)   Fourth root of the fourth moment

$$\text{FRF}M_{i,r} = \sqrt[4]{E(xs_{i,r} - x\bar{s}_{i,r})^4} \tag{17}$$

where $x\bar{s}_{i,r}$ is the mean value of $xs_{i,r}$.

2.2.6. Target Vertical Region Recognition

After feature extraction from all segmented A-scan signals, another BP neural network is used as the classifier to recognize each segment to obtain the vertical regions of targets. The BP neural network also consists of three layers. The input layer contains three neurons, the hidden layer contains ten neurons, and the output layer contains two neurons.

The procedure of target vertical region recognition based on the BP neural network is summarized as follows:

1.   Training set construction. Km2 segments with target reflections and kn2 segments without target reflections are selected from the segmented A-scan signals for training, and three features of each selected segment are extracted. Then, the features are normalized to construct the training set TR2, including km2 positive samples and kn2 negative samples. The output of the positive sample is set to [1 0], and the output of the negative sample is set to [0 1].

2.   Network training. The training set TR2 is used to train the BP neural network, and the network model NET2 is obtained.

3.   Vertical region recognition. The three features of all segments in the optimized horizontal regions $H3$ of the test B-scan image are extracted and normalized to form the test set TE2. Then, the model NET2 is used to classify the samples in TE2. Assuming that $J_i$ samples are identified as positive samples in the A-scan signal $x_i$ ($i = i_1, i_2, \cdots, i_{K3}$), the corresponding segments can be written as $xs_{i,r}$ ($r = r_1, r_2, \cdots, r_{J_i}$). Then, the target vertical regions in the A-scan signal $x_i$ can be denoted as $V1_i = \left\{ \left[ ((r_p - 1)ml + 1)\Delta t, r_p ml \Delta t \right], 1 \le p \le J_i \right\}$.

4.   The recognized segments are arranged in the two-dimensional image, and then the final target regions can be obtained.

## 3. Results

To evaluate the proposed target detection method, both numerical simulations and field experiments are carried out. The simulation data are generated using the "gprMax" simulator based on the finite-difference time-domain (FDTD) method [32], and the real field data are obtained from a road evaluation using a commercial impulse GPR system with a central frequency of 400 MHz. All the programs are executed on a 3.60-GHz CPU and 16 GB-memory computer.

*3.1. Numerical Simulations*

Figure 6 shows the geometry of the simulation model. The model has a depth of 1.6 m and a width of 12.0 m. The model consists of three layers: 0.1 m thick air ($\varepsilon_r = 1$, $\sigma = 0$), 0.2 m thick concrete ($\varepsilon_r = 6$, $\sigma = 0.01$), and 1.3 m thick soil ($\varepsilon_r = 8$, $\sigma = 0.003$). The model contains three targets: one circular void, one metal pipe ($\varepsilon_r = 300$, $\sigma = 10^8$), and one PVC pipe ($\varepsilon_r = 3$, $\sigma = 0.01$). The radius of the void is 0.1 m, the outer radius of the two pipes is 0.1 m, and the inner radius of the two pipes is 0.05 m. The inside of two pipes is filled with water ($\varepsilon_r = 81$, $\sigma = 0.001$). The three targets are buried at the same depth of 0.4 m. The lateral distances of the three targets are 3.0 m, 6.0 m, and 9.0 m, respectively.

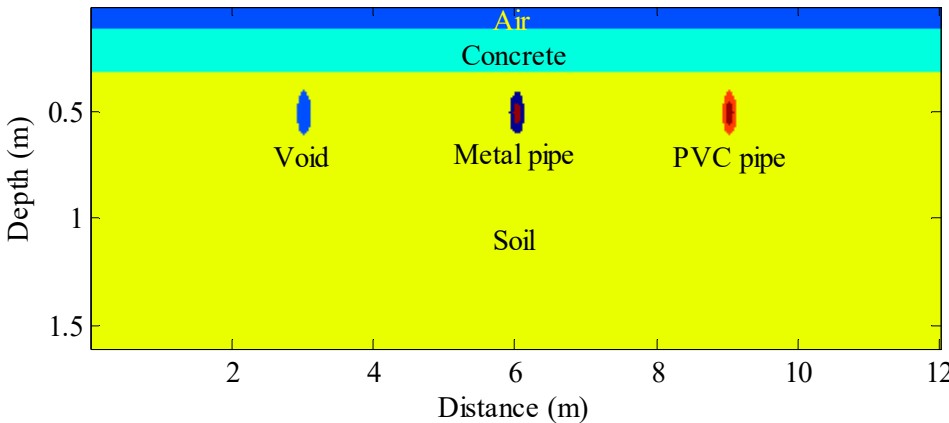

**Figure 6.** Geometry of the simulation model.

The parameters of FDTD simulation are listed in Table 1. To verify the performance of RPCA, noise and clutter are added to the original GPR image. First, Gaussian white noise is added to the original GPR image. Then, the low-frequency components of three A-scan signals with target reflections are added to all A-scan signals as horizontal clutter. Finally, five small regions containing target reflections extracted from the image are added to different positions in the image as the point clutter. Figure 7 shows the simulated GPR image.

**Table 1.** Parameters of FDTD simulation.

| Parameter | Value |
| --- | --- |
| Antenna central frequency | 400 MHz |
| Excitation waveform | Ricker wavelet |
| Time window | 20 ns |
| Number of time samples | 848 |
| Trace interval | 0.02 m |
| Number of traces | 590 |

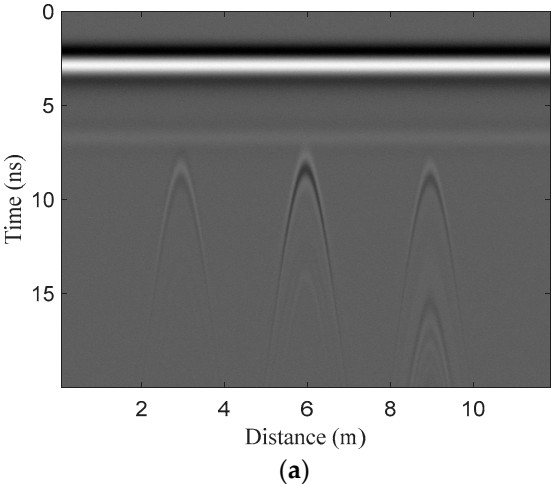

(**a**)

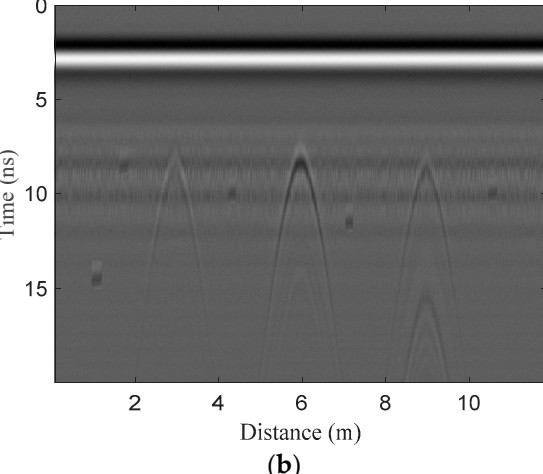

(**b**)

**Figure 7.** Simulated GPR images: (**a**) original image; (**b**) image with clutters and noise.

The signal-to-clutter ratio (SCR) [33] is used to measure the quality of the GPR image, which is defined as

$$SCR = \frac{N_c \sum\limits_{p \in R_t} |I(p)|^2}{N_t \sum\limits_{p \in R_c} |I(p)|^2} \tag{18}$$

where $I(p)$ is the $p$-th pixel in the image, $R_t$ is the target region, $R_c$ is the clutter region, and $N_c$ and $N_t$ are the number of pixels in the clutter and target regions, respectively. The target region is indicated by a box containing the target, and the clutter region is defined as the entire image excluding the target region.

As shown in Figure 7, the strong direct wave almost masks the effective reflection signals of targets, and the SCR values of the original image and the image with clutter and noise are $-10.3$ dB and $-12.5$ dB, respectively. Because the ratio of the direct wave energy to total image energy is very large, the difference in SCR between the two images is not obvious.

First, RPCA is applied to the simulated image with clutters and noise, and the experimental results of RPCA are compared with those of PCA. Figure 8 shows the clutter suppression results of the two methods. As shown in Figure 8a, PCA eliminates the direct wave, but it retains the point clutter, some horizontal clutter, and some noise. As shown in Figure 8b, RPCA can remove the direct wave and horizontal clutter completely, but it also retains the point clutter and some noise. The SCR of PCA and RPCA is 4.4 dB and 6.6 dB, respectively. The results show that RPCA has better clutter suppression performance than PCA, especially for horizontal clutter. Similar to target reflection signals, the point clutters in the image are sparsely distributed, so they cannot be removed by the two methods. The noise still exists in the sparse matrix of RPCA and the target component of PCA after the decomposition, so the two methods are not available for noise suppression.

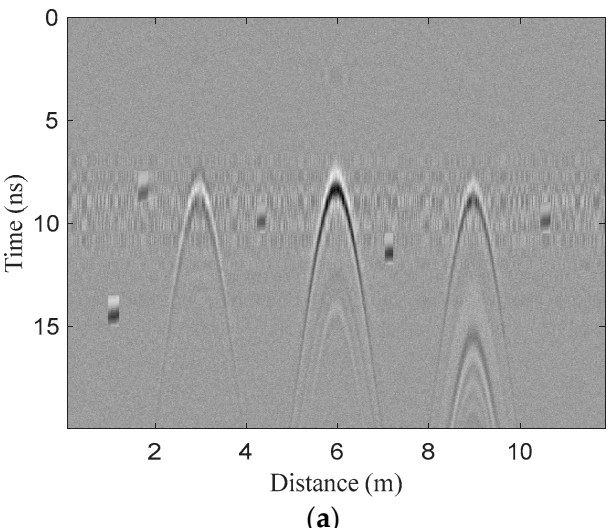

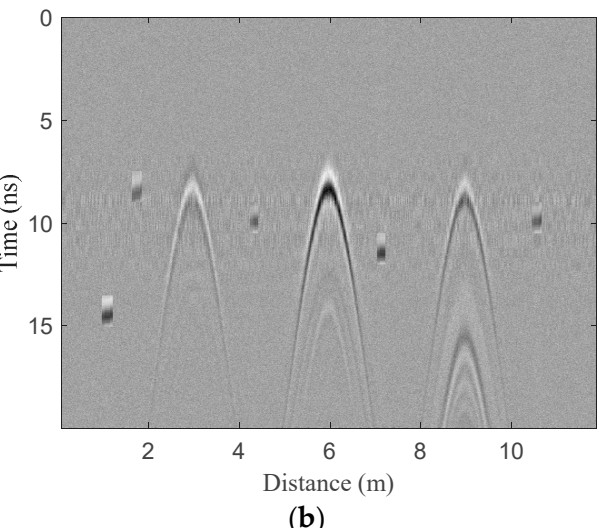

**Figure 8.** Clutter suppression results of two methods for the simulated GPR image: (**a**) PCA (SCR = 4.4 dB); (**b**) RPCA (SCR = 6.6 dB).

In the simulation model, the main parameters of the underground target include depth, lateral distance, and radius. Nine simulation models are established by setting different target parameters, and nine simulated GPR images are obtained from the nine models. Table 2 lists the parameters of the three targets in the nine models.

Then, 400 A-scan signals with target reflections and 400 A-scan signals without target reflections are selected from the nine simulated images after clutter suppression. The three features of the 800 A-scan signals are extracted to construct the training set TR1, in which each sample is a vector with three features. Similarly, the three features of all A-scan signals in the test image in Figure 8b are extracted to construct the test set TE1.

**Table 2.** Parameters of the three targets for different simulation models.

| Model Number | Void | | | Metal Pipe | | | PVC Pipe | | |
|---|---|---|---|---|---|---|---|---|---|
| | Depth | Lateral Distance | Radius | Depth | Lateral Distance | Radius (Outer/Inner) | Depth | Lateral Distance | Radius (Outer/Inner) |
| 1 | 0.30 m | 3 m | 0.10 m | 0.30 m | 6 m | 0.10 m/0.05 m | 0.30 m | 9 m | 0.10 m/0.05 m |
| 2 | 0.50 m | 3 m | 0.10 m | 0.50 m | 6 m | 0.10 m/0.05 m | 0.50 m | 9 m | 0.10 m/0.05 m |
| 3 | 0.70 m | 3 m | 0.10 m | 0.70 m | 6 m | 0.10 m/0.05 m | 0.70 m | 9 m | 0.10 m/0.05 m |
| 4 | 0.35 m | 3 m | 0.15 m | 0.35 m | 6 m | 0.15 m/0.10 m | 0.35 m | 9 m | 0.15 m/0.10 m |
| 5 | 0.55 m | 3 m | 0.15 m | 0.55 m | 6 m | 0.15 m/0.10 m | 0.55 m | 9 m | 0.15 m/0.10 m |
| 6 | 0.75 m | 3 m | 0.15 m | 0.75 m | 6 m | 0.15 m/0.10 m | 0.75 m | 9 m | 0.15 m/0.10 m |
| 7 | 0.40 m | 3 m | 0.20 m | 0.40 m | 6 m | 0.20 m/0.15 m | 0.40 m | 9 m | 0.20 m/0.15 m |
| 8 | 0.60 m | 3 m | 0.20 m | 0.60 m | 6 m | 0.20 m/0.15 m | 0.60 m | 9 m | 0.20 m/0.15 m |
| 9 | 0.80 m | 3 m | 0.20 m | 0.80 m | 6 m | 0.20 m/0.15 m | 0.80 m | 9 m | 0.20 m/0.15 m |

Figure 9 shows the three features of all A-scan signals in the test image, which are the samples in test set TE1. As shown in Figure 9, the amplitude of the three features increases significantly in the horizontal regions of three targets, as well as in the horizontal regions of the five-point clutter regions. The results show that the three features can distinguish target reflections and non-target reflections but fail to distinguish target reflections and point clutter reflections.

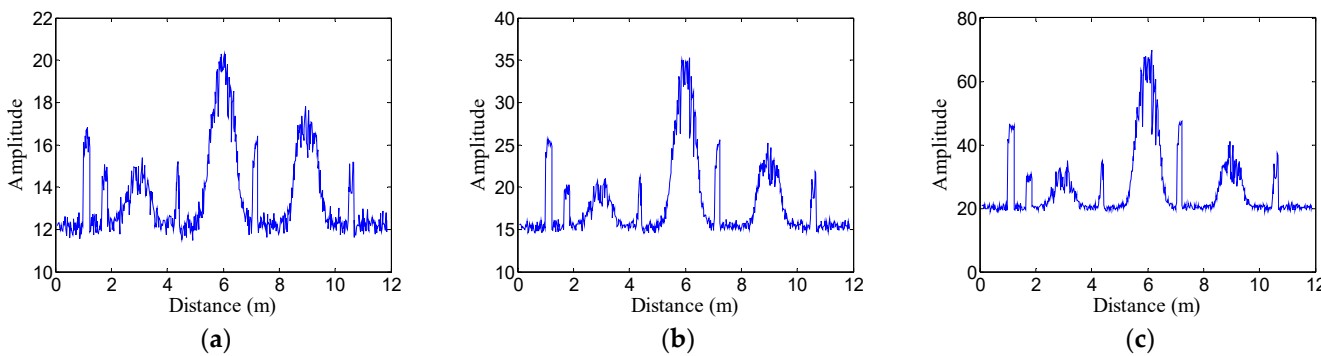

**Figure 9.** Three features of all A-scan signals in the test image: (**a**) mean absolute deviation; (**b**) standard deviation; (**c**) fourth root of the fourth moment.

The training set TR1 is used to train one BP neural network with the structure of 3-10-2 (number of neurons in the input-layer–hidden-layer–output-layer) to obtain a network model NET1, which is used to recognize the samples in the test set TE1 to obtain the target horizontal regions. Then, the FAD algorithm is used to further optimize the recognized horizontal regions. The horizontal interval threshold *dth* is determined according to the target region width in the features of A-scan signals in the nine training images, which is generally 1/10 of the average width of the target region. The horizontal width threshold *wth* is determined according to the clutter region width in the features of A-scan signals in the training images, which is generally twice the maximum width of the clutter region. Here, *dth* and *wth* are set to 0.1 m and 0.3 m, respectively.

Figure 10 shows the recognized original target horizontal regions and optimized target horizontal regions. As shown in Figure 10a, the original target horizontal regions contain multiple regions. Five-point clutter regions have a width of less than 0.3 m, which represents false detection. In addition, there are several intervals with widths less than 0.1 m on the sides of the three target regions, which represent the missing detection. As shown in Figure 10b, the FAD algorithm can effectively overcome false detection and missing detection and obtain more accurate horizontal regions for the three targets.

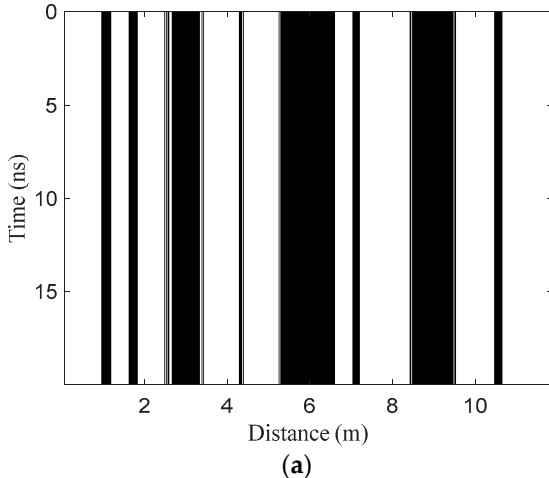 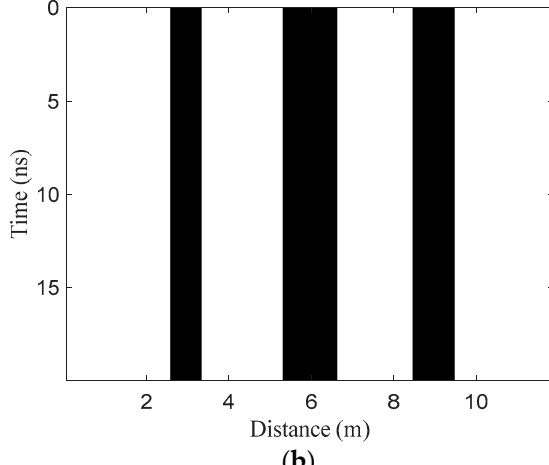

(**a**)   (**b**)

**Figure 10.** Target horizontal region recognition results for the simulated GPR image: (**a**) original horizontal regions; (**b**) optimized horizontal regions.

Here, three metrics, accuracy, false positive rate (FPR), and false negative rate (FNR) are used to quantitatively measure the horizontal region recognition performance of the proposed method. The three metrics are given by

$$Accuracy = \frac{TP + TN}{TP + TN + FP + FN} \tag{19}$$

$$FPR = \frac{FP}{FP + TN} \tag{20}$$

$$FNR = \frac{FN}{TP + FN} \tag{21}$$

where *TP* is the number of targets that are correctly classified, *TN* is the number of non-targets that are correctly classified, *FN* is the number of targets that are incorrectly classified, and *FP* is the number of non-targets that are incorrectly classified.

The accuracy indicates the proportion of correctly classified samples to the total samples, which reflects the overall recognition performance. FPR indicates the proportion of incorrectly classified non-targets to the total non-targets, which reflects the false-detection performance. FNR indicates the proportion of incorrectly classified targets to the total targets, which reflects the missing detection performance.

Figure 11 shows the confusion matrices of target horizontal region recognition before and after optimization with the FAD algorithm. Table 3 shows the three metrics before and after optimization with the FAD algorithm. It can be seen that after optimization, the accuracy is improved by 9.5%, FPR is reduced by 10.8%, and FNR is reduced by 6.1%. The results show that the FAD algorithm can effectively improve the recognition performance of target horizontal regions.

**Table 3.** Recognition performance of target horizontal regions for the simulated GPR image.

| Method | Accuracy | FPR | FNR |
| --- | --- | --- | --- |
| BP neural network | 87.6% | 13.8% | 8.6% |
| BP neural network + FAD | 97.1% | 3.0% | 2.5% |

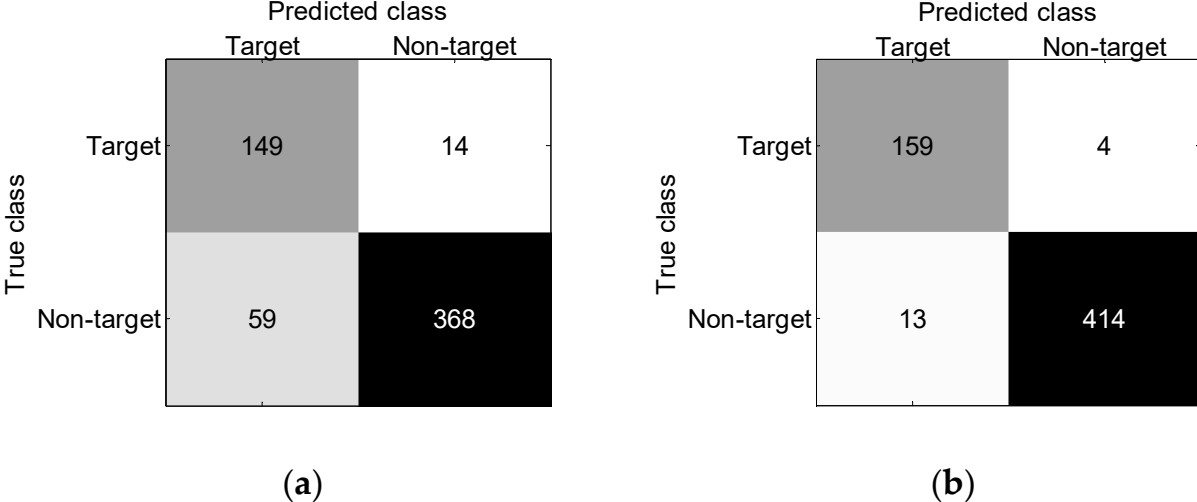

(**a**)

(**b**)

**Figure 11.** Confusion matrices of target horizontal region recognition for the simulated GPR image: (**a**) original horizontal regions; (**b**) optimized horizontal regions.

Finally, 800 previously selected A-scan signals are segmented. Considering the size of the target hyperbola in the vertical direction, the segment length is set to 64. Thus, each A-scan signal is divided into 13 segments, and a total of 10,400 segments are obtained. The three features of the 10,400 segments are extracted to construct the training set TR2. Similarly, all 163 A-scan signals in the optimized horizontal regions are also segmented, and the total number of segments is 2119. The three features of the 2119 segments are extracted to construct the test set TE2.

Figure 12 shows the three features of segments in one A-scan signal in the horizontal region of the PVC pipe, which are also the samples in the test set TE2. As shown in Figure 12, the amplitude of the three features of the segments with target reflections is higher than that of the segment without target reflections, which shows that the three features are also applicable to the segmented A-scan signals.

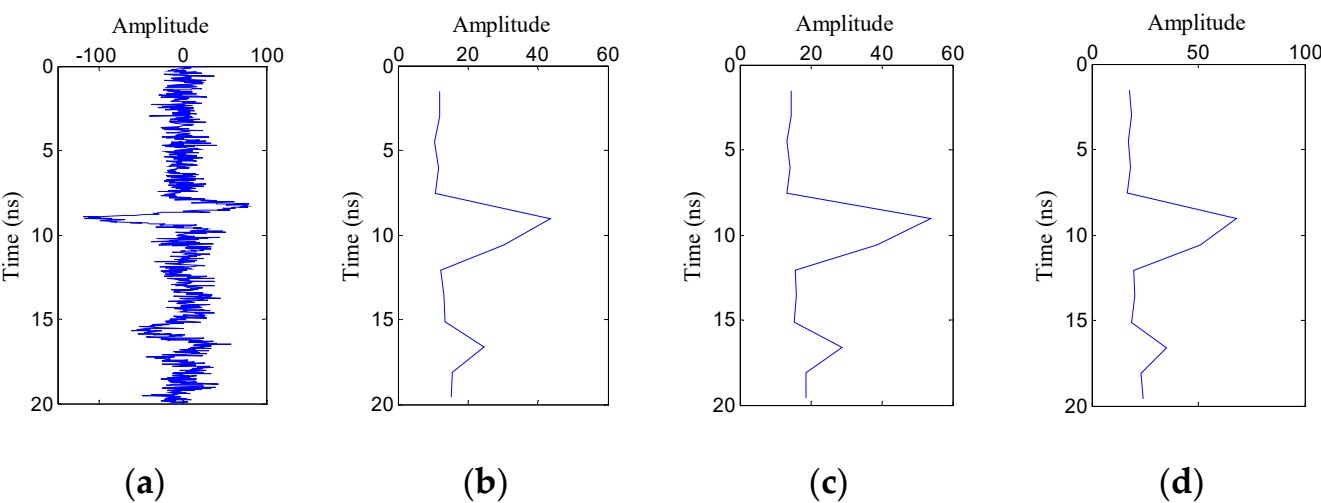

(**a**)　　　　　　　(**b**)　　　　　　　(**c**)　　　　　　　(**d**)

**Figure 12.** Three features of segments in one A−scan signal: (**a**) A−scan signal; (**b**) mean absolute deviation; (**c**) standard deviation; (**d**) fourth root of the fourth moment.

The training set TR2 is used to train one BP neural network with the structure of 3-10-2 to obtain another network model NET2, which is used to recognize the samples in the test set TE2 to obtain the target vertical regions. The target regions are obtained from the recognized target segments in the vertical regions.

For comparison, the traditional segmentation recognition methods based on the BP neural network [9] and SVM [13] are also adopted to process the image in Figure 8b. The method based on the BP neural network uses twelve spectral values of segments as features, and the method based on SVM uses three time-domain statistics of segments as features.

Figure 13 shows the target recognition results of the three methods. As shown in Figure 13a,b, some point clutter regions are identified as target reflections, and several clutter regions near the hyperbolas of targets are also identified as target reflections. As shown in Figure 11c, the hyperbolic reflections of targets are identified clearly, and only a few clutter regions are recognized as target reflections. The results show that the proposed method has better target recognition performance than traditional segmentation recognition methods.

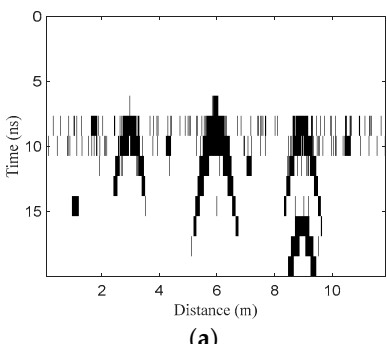 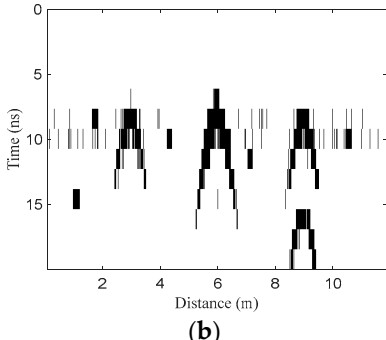 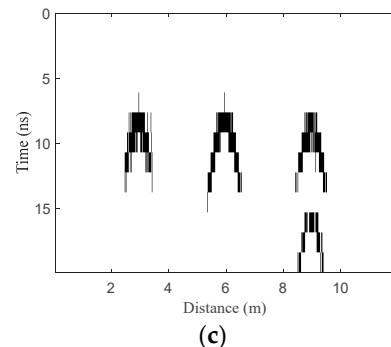

(**a**) (**b**) (**c**)

**Figure 13.** Target recognition results for the simulated GPR image: (**a**) traditional segmentation recognition method based on BP neural network; (**b**) traditional segmentation recognition method based on SVM; (**c**) proposed method.

Figure 14 shows the confusion matrices of the three methods. Table 4 lists the three metrics of the three methods. The FPR of the proposed method is much lower than that of the other two methods, which shows that the proposed method can greatly reduce false detection. However, the FNR of the proposed method reaches 18.9%, which indicates that the proposed method still has a small amount of missing detection in the recognition of target vertical regions. Because the number of target segments is much lower than the number of non-target segments in the GPR image, the influence of FNR on the recognition accuracy is much less than that of FPR, which also explains why the proposed method can achieve higher accuracy than the other two traditional methods. As shown in Figure 11c, the small amount of missing target reflections will not affect the final detection of target regions.

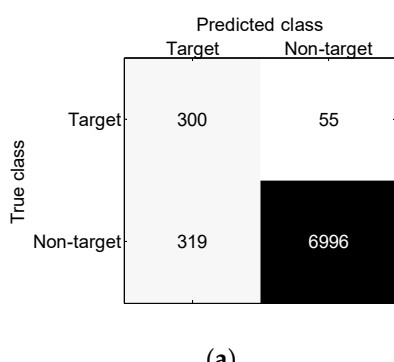 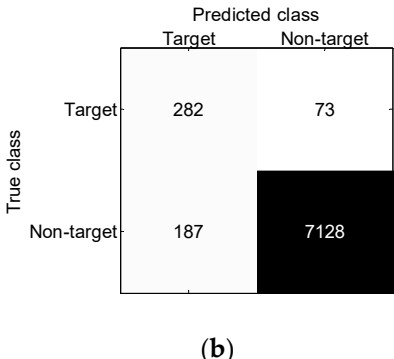 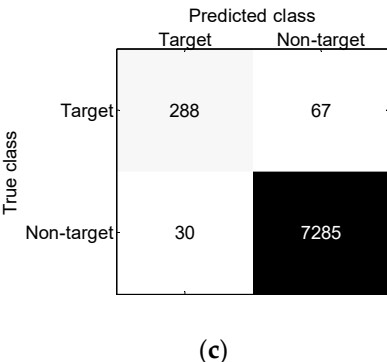

(**a**) (**b**) (**c**)

**Figure 14.** Confusion matrices of the three methods for the simulated GPR image: (**a**) traditional segmentation recognition method based on BP neural network; (**b**) traditional segmentation recognition method based on SVM; (**c**) proposed method.

**Table 4.** Target recognition performance of the three methods for the simulated GPR image.

| Method | Accuracy | FPR | FNR |
|---|---|---|---|
| Traditional segmentation recognition method based on BP neural network | 95.1% | 4.4% | 15.5% |
| Traditional segmentation recognition method based on SVM | 96.6% | 2.6% | 20.6% |
| Proposed method | 98.7% | 0.4% | 18.9% |

Table 5 lists the processing times of the three methods. The processing time refers to the time consumption of processing the test image, which includes a clutter suppression stage and a target-recognition stage. The clutter suppression time of the three methods is the same. The target recognition time of the traditional two methods consists of the construction and classification time of the test set. The target recognition time of the proposed method consists of the construction and classification time of the test set TE1 (horizontal region recognition time), the horizontal region optimization time, and the construction and classification time of the test set TE2 (vertical region recognition time). Compared with the two traditional methods, the target recognition time and total processing time of the proposed method are reduced by about 30% and 20%, respectively. The results show that the proposed method achieves higher detection efficiency than the two traditional methods.

**Table 5.** Processing time of the three methods for the simulated GPR image.

| Method | Processing Time (s) | | |
|---|---|---|---|
| | Clutter Suppression | Target Recognition | Total |
| Traditional segmentation recognition method based on BP neural network | 0.38 | 0.63 | 1.01 |
| Traditional segmentation recognition method based on SVM | 0.38 | 0.65 | 1.03 |
| Proposed method | 0.38 | 0.44 | 0.82 |

*3.2. Field Experiments*

A field experiment is conducted with the GPR system on a road in Wuhan (China). The space sampling step length (trace interval) is 0.05 m. The time window is 90.74 ns, and the number of time samples is 512. Figure 15a shows one original B-scan image containing 551 A-scan signals. The image contains the reflections of two targets, the direct wave, horizontal clutter, and a lot of irregular clutter.

First, RPCA is used to suppress the clutter in the original image, and the result is shown in Figure 15b. As shown in Figure 15b, the direct wave is completely removed, most horizontal clutter and irregular clutter is also eliminated, and the reflections of the two targets are well preserved. After clutter suppression, the SCR is improved by 16.9 dB. The result also demonstrates the excellent clutter-suppression performance of RPCA for the real GPR image.

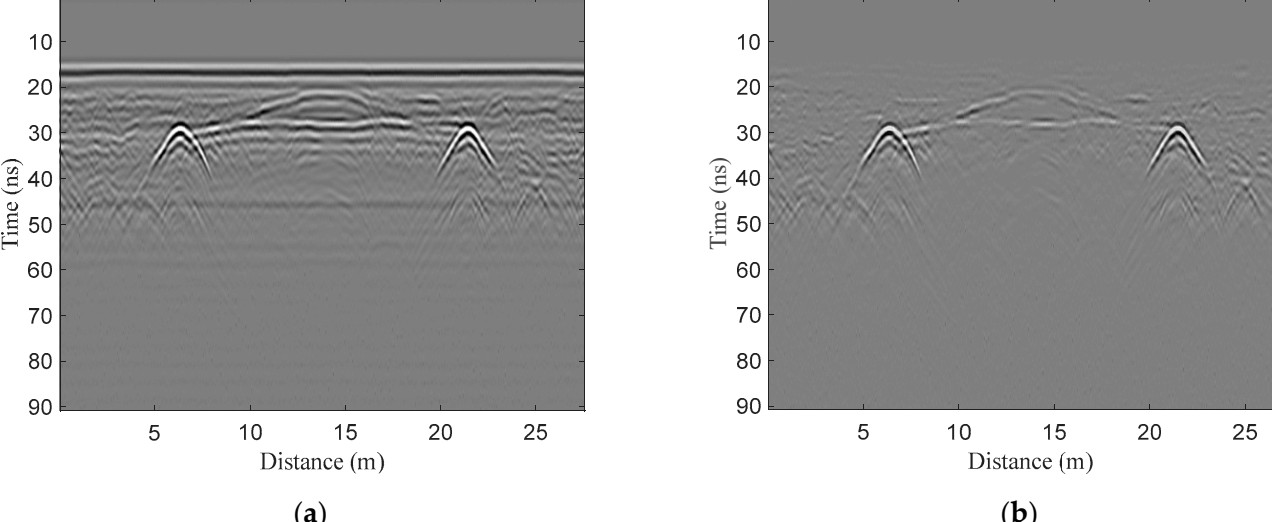

**Figure 15.** Real GPR image: (**a**) original image (SCR = 0.6 dB); (**b**) image after clutter suppression (SCR = 17.5 dB).

Then, 200 A-scan signals with target reflections and 200 A-scan signals without target reflections are selected from 25 real B-can images after clutter suppression. The three features of the 400 A-scan signals are extracted to construct the training set TR1, and the three features of all the A-scan signals in the test image in Figure 15b are extracted to construct the test set TE1. The network model NET1 is obtained by training a neural network with the structure of 3-10-2 using the training set TR1, which is used to recognize the samples in TE1 to obtain the target horizontal regions. Subsequently, the recognized horizontal regions are optimized by the FAD algorithm, and the two parameters *dth* and *wth* are set to 0.1 m and 0.4 m, respectively.

Figure 16 shows the target horizontal region recognition results. As shown in Figure 16a, the original target horizontal regions contain two target regions with larger width and several clutter regions with smaller width. As shown in Figure 16b, the FAD algorithm eliminates the clutter regions and preserves the two target regions well.

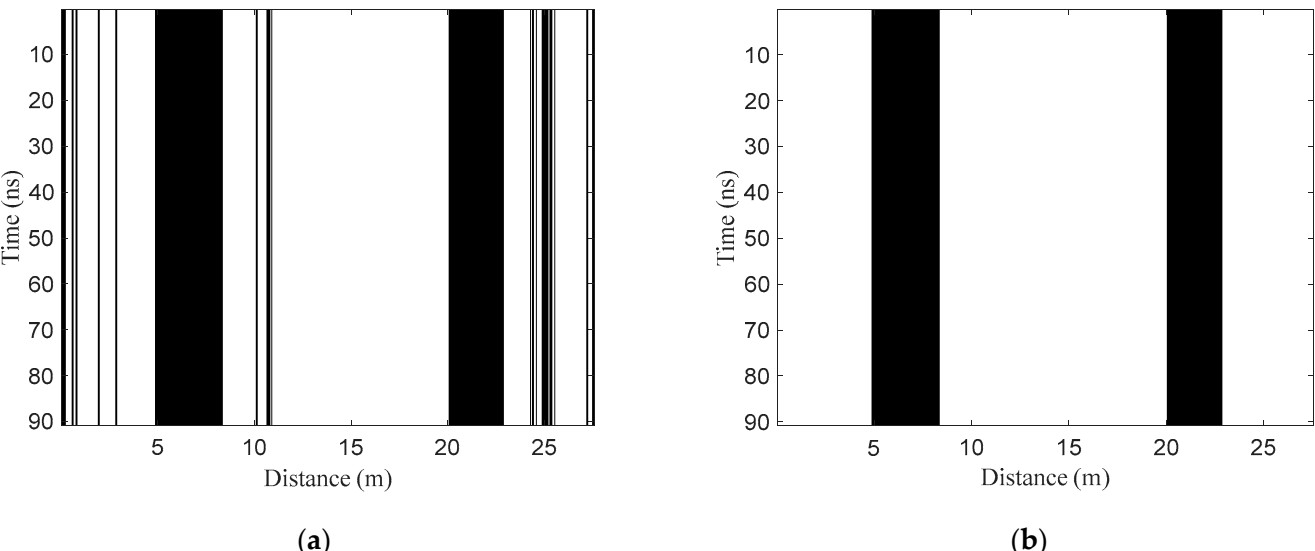

**Figure 16.** Target horizontal region recognition results for the real GPR image: (**a**) original horizontal regions; (**b**) optimized horizontal regions.

Figure 17 shows the confusion matrices of target horizontal region recognition before and after optimization with the FAD algorithm. Table 6 lists the recognition performance of target horizontal regions before and after optimization with the FAD algorithm. It can be seen that after optimization, the accuracy is improved by 7.3% and FPR is reduced by 10%, but FNR remains unchanged. The results again show that the FAD algorithm can effectively improve the accuracy and decrease FPR for recognition of the target horizontal region.

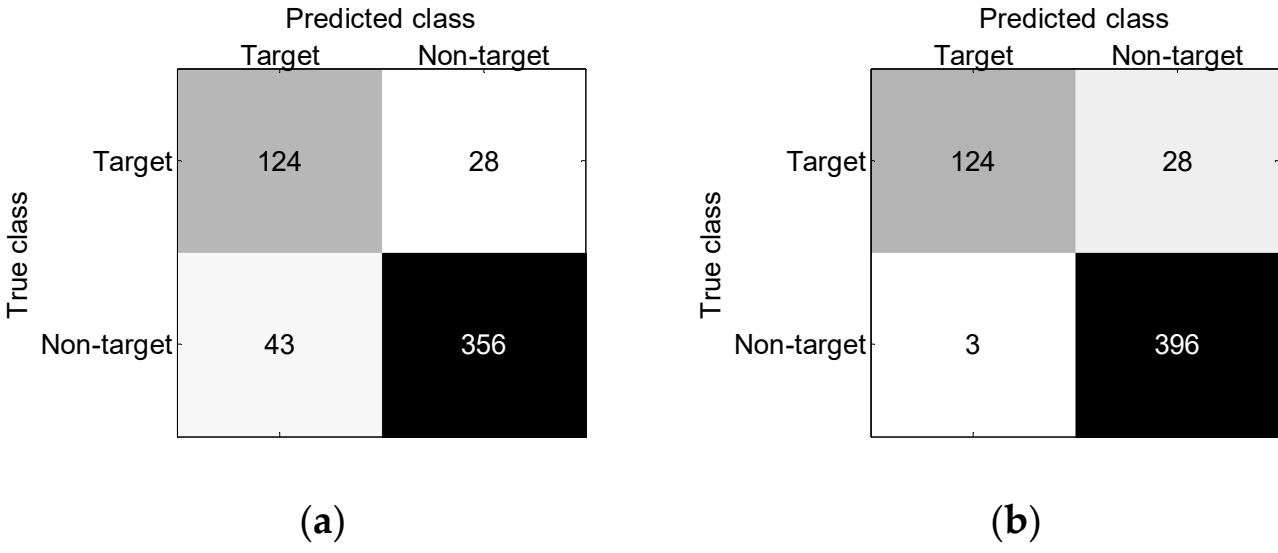

**(a)** **(b)**

**Figure 17.** Confusion matrices of target horizontal region recognition for the real GPR image: (**a**) original horizontal regions; (**b**) optimized horizontal regions.

**Table 6.** Recognition performance of target horizontal regions for the real GPR image.

| Method | Accuracy | FPR | FNR |
|---|---|---|---|
| BP neural network | 87.1% | 10.8% | 18.4% |
| BP neural network + FAD | 94.4% | 0.8% | 18.4% |

Finally, 400 previously selected A-scan signals are segmented, and the segment length is 16. Thus, each A-scan signal is divided into 32 segments, and a total of 12,800 segments are obtained. The three features of the 12,800 segments are extracted to construct the training set TR2. The A-scan signals in the optimized horizontal regions are segmented, and the three features of the segments are extracted to construct the test set TE2. The network model NET2 is obtained by training another neural network with a structure of 3-10-2 using TR2. The target vertical regions are obtained by using NET2 to recognize the samples in TE2, and the recognized target segments in the vertical regions represent the target regions. Here, the experimental results of the proposed method are also compared with those of traditional segmentation recognition methods.

Figure 18 shows the target-recognition results of the three methods. As shown in Figure 18a,b, the two traditional methods generate several wrong misjudgments due to the influence of residual clutter, and the two targets can hardly be identified. As shown in Figure 18c, the proposed method can clearly identify the two targets with few misjudgments. The results also show that the proposed method is superior to traditional segmentation recognition methods.

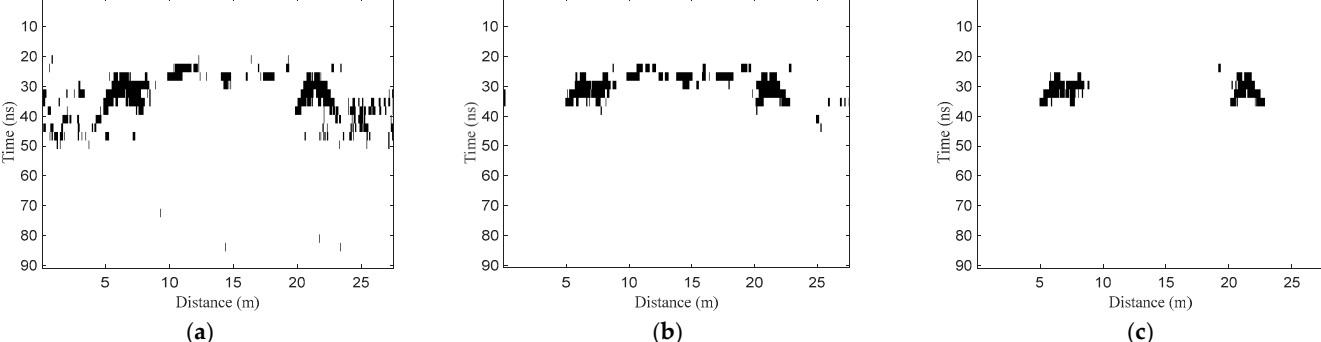

**Figure 18.** Target recognition results for the real GPR image: (**a**) traditional segmentation recognition method based on BP neural network; (**b**) traditional segmentation recognition method based on SVM; (**c**) proposed method.

Figure 19 shows the confusion matrices of the three methods. Table 7 lists the three metrics of the three methods. The FPR of the proposed method is also much lower than that of the two traditional methods, but the FNR of the proposed method is slightly higher than that of the two traditional methods. The results show that the proposed method greatly reduces the false detection but slightly increases the missing detection, which is also consistent with the recognition results in Figure 19. As FPR is greatly reduced, the proposed method still achieves higher accuracy than the two traditional methods.

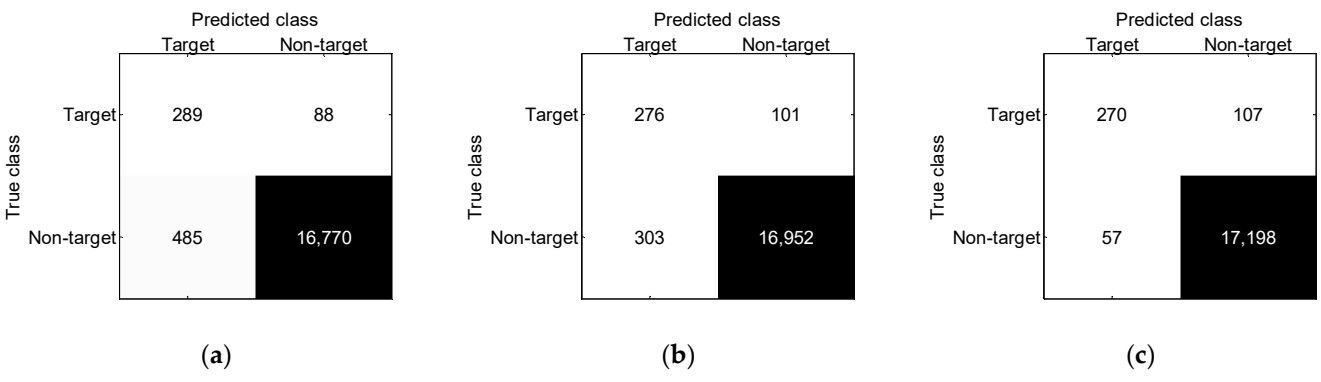

**Figure 19.** Confusion matrices of the three methods for the real GPR image: (**a**) traditional segmentation recognition method based on BP neural network; (**b**) traditional segmentation recognition method based on SVM; (**c**) proposed method.

**Table 7.** Target recognition performance of the three methods for the real GPR image.

| Method | Accuracy | FPR | FNR |
|---|---|---|---|
| Traditional segmentation recognition method based on BP neural network | 96.8% | 2.8% | 23.3% |
| Traditional segmentation recognition method based on SVM | 97.7% | 1.8% | 26.8% |
| Proposed method | 99.1% | 0.3% | 28.4% |

Table 8 lists the processing time of the three methods. Compared with the two traditional methods, the target-recognition time and total processing time of the proposed

method are reduced by about 20% and 14%, respectively. The results also show that the proposed method is more efficient than the two traditional methods.

**Table 8.** Processing time of the three methods for the real GPR image.

| Method | Processing Time (s) | | |
|---|---|---|---|
| | Clutter Suppression | Target Recognition | Total |
| Traditional segmentation recognition method based on BP neural network | 0.54 | 0.93 | 1.47 |
| Traditional segmentation recognition method based on SVM | 0.54 | 0.91 | 1.45 |
| Proposed method | 0.54 | 0.74 | 1.28 |

## 4. Discussion

This paper proposes an efficient GPR target-detection method based on horizontal and vertical region recognition using a BP neural network. Preprocessing is executed using RPCA, horizontal region recognition is based on the recognition of A-scan signals, and vertical region recognition is based on the recognition of segments in A-scan signals. The proposed two-stage recognition structure can reduce the recognition of segments of A-scan signals in non-target regions in traditional methods. In order to better describe the reflection characteristics of the target, three time-domain statistics are selected as features. In addition, a simple FAD algorithm is proposed to optimize the horizontal region.

A series of simulated and real GPR data was used to verify the proposed method. The results show that PRCA can effectively suppress non-sparse clutter, but it is not suitable for sparse clutter and noise suppression. In the recognition of the simulated image in the horizontal region, the three features can distinguish target regions and non-target regions but cannot distinguish target regions and point clutter regions. The FAD algorithm can effectively reduce the influence of point clutter and improve the recognition accuracy of horizontal regions. The proposed method is also compared with two traditional segmentation recognition methods. The comparison results show that the proposed method can significantly reduce FPR and improve accuracy but cannot reduce FNR. In addition, the results also show that the proposed method can effectively reduce the processing time.

## 5. Conclusions

In this paper, an efficient recognition method based on neural networks is proposed to improve the detection performance of underground road targets in GPR images. RPCA is first used to suppress the clutter in the image. Then, one BP neural network is adopted to obtain the horizontal regions of targets by recognizing A-scan signals in the image, and another BP neural network is used to obtain the vertical regions of targets by recognizing segmented A-scan signals in the horizontal regions of targets, which provides a solution to improve the recognition efficiency. Moreover, the FAD algorithm is presented to optimize the horizontal regions of targets.

The effectiveness of the proposed method is verified by both simulated and real GPR images. The experimental results show that the proposed method is superior to the two traditional segmentation recognition methods in recognition accuracy and processing time. Future work will study more efficient target recognition solutions to improve GPR detection performance.

**Author Contributions:** Conceptualization, W.X. and K.C.; methodology, W.X.; software, K.C.; validation, K.C. and T.L.; formal analysis, W.X. and J.Z.; data curation, K.C.; writing—original draft preparation, K.C.; writing—review and editing, W.X. and J.Z.; funding acquisition, L.L. All authors have read and agreed to the published version of the manuscript.

**Funding:** This work was partially supported by the National Natural Science Foundation of China (62175220).

**Data Availability Statement:** The data presented in the study are available on request from the first author.

**Conflicts of Interest:** The authors declare no conflict of interest.

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
