# Peer review of "Efficient Underground Target Detection of Urban Roads in Ground-Penetrating Radar Images Based on Neural Networks"

_remotesensing, doi:10.3390/rs15051346_

Round 1

Reviewer 1 Report

Dear authors,

thank you very much for this nice work on hyperbola detection in GPR data. I like the relatively simple approach using a very small neural network for detecting horizontal and then vertical regions of interest. Nevertheless, the text could be a bit easier to read by reducing unneccessary parts (see comment in PDF). The essence of your approach became obvious to me only in the results part. Before I thought that it is a very complicated approach, but in the end it is not. Please try to shorten your text and be more concrete about the steps that you are really doing. Sometimes you mention also other approaches, but these are not used. They just confuse the reader. If you can also find the fomulas somewhere else in literature, just refer to them. In my opinion they just blow up the text but are not neccessary to have them directly in the text. The reader can look them up in the literature.

The introduction lists many references from machine learning and deep learning and contains a lot of information. Nevertheless, I'm not sure why you explain so mauhc about deep learning, when you are using machine learning in the end. Also in my opinion when you use a hidden layer you have a deep neural network and thus are doing deep learning. Please clarify the difference between machine learning and deep learning or do not divide it at all.

It would be nice to have some example values of the features that you calculated for different regions of the radargram. Maybe you can also plot the three features as figures?

I started making some suggestions for English correction, but I'm not a native speaker and would highly suggest proof reading by a native speaker.

Reviewer 2 Report

omments and suggestions to authors:

I suggest that the following points might be adressed or corrected:
L154: remove squares for x_n and x_0 in formula 1.
L196: describe s.t. in formula 3.
L218: be more explicite about features selection. They are also applied on no gained data, so without spreading and exponential corrections (SEC). Whi not try different statistics parameters, based for instance on STA/LTA ratios, or AGC-type values.
We could also think that absolute values of x could be more appropriate (after  DC-shift corrrection), or a processing on signal envelop afte Hilbert transform calculation.
L232: recognizing hyperbolic signatures on A-scan does not seem reasonable: only B-scan reveal such 2D responses.
L243: could you explain empirical formula 8.
L254: why not use a classical ReLU function, instead of purelin?
L321: how is "ml" value selected ? Does it depend on central GPR frequencu, shifted from nominal value with attenuation parameters?
L369: Computer performances are quite useless as the modelling process is not so heavy, and the BP on a 3 layers NN is quite computationnaly painless.
L386: please explicit the adding noise process, as the point clutters distribution in the image is discussed (L413). Nevertheless, we do not understand while 5 punctual reflection appear at exactly the same depth as the void or pipes depth. (Figure 8b).
L425: we suppose that the 400 + 400 A-scan signals are selected among 9 models x 590 traces per B-scan.
L434: one again, please explict threshold value selection.
L447: it is also  quite usual for machine learning to show the confusion matrix.
L484-494: please (re-?)reference (a) and (b) processing in Figure 11 (or in Figure 14).
L538: we are quite surprised that the stratigraphic signature (even with learning) does not generate an equivalent answer.
